# Building System Capacity with a Modeling-Based Inquiry Program for Elementary Students: A Case Study



Amanda M. Cottone [1,*], Susan A. Yoon [1], Bob Coulter [2], Jooeun Shim [1] and Stacey Carman [2]

1   Graduate School of Education, University of Pennsylvania, Philadelphia, PA 19014, USA; yoonsa@upenn.edu (S.A.Y.); jshim@upenn.edu (J.S.)
2   Litzsinger Road Ecology Center, Missouri Botanical Gardens, Ladue, MO 63124, USA; bob@lrec.net (B.C.); carmans@swbell.net (S.C.)
*   Correspondence: amandaco@upenn.edu

**Abstract:** Science education in the United States should shift to incorporate innovative technologies and curricula that prepare students in the competencies needed for success in science, technology, engineering, and math (STEM) careers. Here we employ a qualitative case study analysis to investigate the system variables that supported or impeded one such reform effort aimed at improving elementary students' science learning. We found that, while some program design features contributed to the success of the program (i.e., a strong multi-institutional partnership and a focus on teacher training and instructional supports), other features posed barriers to the long-term system-level change needed for reform (i.e., low levels of social capital activation, low prioritization of science learning, and frequent turnover of key personnel). In light of these findings, we discuss broader implications for building the capacity to overcome system barriers. In this way, an in-depth examination of the context-specific barriers to reform in this educational system can inform efforts for future reform and innovation design.

**Keywords:** data literacy; elementary education; agent-based simulations; StarLogo Nova; science curriculum; complex systems learning; usability cube; student engagement

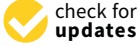



## 1. Introduction

"What works, for whom, and under what conditions" has been a mantra in educational research in order to address the issue of enacting high-quality and successful educational reform [1–3]. The need for this approach has never been truer in science, technology, engineering, and mathematical (STEM) education, where continued workforce shortages and the underrepresentation of women and minorities in STEM occupations in the United States necessitate effective, long-lasting, and extensive science education reform [4,5]. Young learners are too often overlooked in such reform efforts [6], yet it is important to begin integrated and impactful STEM education as early as possible in order to spark learners' interest and better prepare and train them in the competencies required for future success. Data literacy skills and model-based learning are two essential components to STEM education that have gained some traction at the elementary level [7,8], but overall, these components are lacking in any substantive way in elementary science curricula [9,10]. One reason for this absence is that teachers need significant support in enacting effective instruction in early science education [3,11]. In addition, systemic barriers challenge the widespread usability of reforms [12,13]. In this study, we examine the case of a model-based inquiry program developed to address these issues, and we use a system-level approach to understand the successes and challenges that influenced the reform effort. We interviewed key players involved in the multi-institutional, districtwide program to uncover the specific supports and hindrances affecting the extent to which the program was taken up by the school district [14]. In conducting this research, our focus extends beyond classroom impact and incorporates the school system as the primary unit of analysis. In this way,

we intend to contribute to the corpus of research that aims to identify the practices and features needed for sustainability (i.e., existing beyond the time the researcher is in the classroom) and scalability (i.e., spreading beyond participating teachers and schools) of important early science education reforms. Additionally, we intend to demonstrate the investigatory resources that can be used to understand the system-level variables impacting the success of a reform effort, which in turn will help us tailor future reform efforts to local conditions. We were guided by the following research questions: What were the program features that either supported or hindered the sustainable, districtwide implementation of a model-based inquiry curriculum for elementary students? What were the affordances to students who participated in the program?

### 1.1. Data Literacy Skills and Model-Based Learning as Important Components of Early STEM Education

Students enter the STEM pipeline in elementary school, yet this early developmental point is often overlooked as an opportunity to counteract the low recruitment and low diversity trends that are evident in the STEM workforce [4–6]. This oversight occurs even though research has highlighted the importance of elementary-level education as one of many critical points in influencing an individual's commitment to pursuing occupations in STEM [15]. In fact, early exposure to high-quality instruction in math and science can stimulate interest in and instill favorable attitudes toward STEM learning that later become key predictors of retention [15,16].

The Next Generation Science Standards (NGSS) in the United States are STEM education guidelines crafted to promote workforce readiness. They emphasize integrating engineering, technology, and real-world science at every grade level of science education [17]. Developing data literacy skills and engaging in model-based learning are two important components highlighted within the NGSS, and both skill sets are featured in the Science and Engineering Practices of the NGSS in, for example, a focus on *Analyzing and Interpreting Data* and *Developing and Using Models* [18].

Data literacy involves being able to understand, create, analyze, and communicate data as information, as well as to question and problematize the ways data are created and used [19]. A number of data literacy skills are essential to introduce to young learners [9]. These include, but are not limited to, understanding the nature and purpose of data through sampling and measurement, noticing trends and patterns within and across datasets, and making inferences based on evidence [9].

Model-based learning refers to the understanding gained from the ability to create, manipulate, and communicate models; to the extent that model-based learning involves being able to collect and use data, it overlaps with the skills needed for data literacy [20]. As an instructional strategy, model-based learning offers a compelling alternative to traditional pedagogy, as well as an opportunity for elementary students to use and engage with sense-making tools that are pervasive in modern science [21]. Instead of memorizing abstract principles learned from texts and formulaic activities, models help students see the integration of STEM concepts that are often taught as standalone ideas. The use of dynamic, agent-based modeling platforms in particular, such as StarLogo Nova, makes complex concepts accessible and inviting to comparatively young students; such platforms also allow learners to work directly with the model by adjusting key parameters and underlying assumptions embedded within it [22]. In addition, these tools can offer unique affordances and support to students as they learn to reason with data [9]. It is, therefore, clear that efforts to immerse elementary students in both data literacy and modeling practices are likely important for strengthening the STEM pipeline and piquing the interest of diverse learners.

While a multitude of small-scale research studies at the classroom level have successfully engaged learners in integrative STEM education, this fragmented approach has yet to result in the more widespread diffusion of such educational innovations [23]. Thus, neither data literacy nor model-based learning is covered in any substantial or sustainable way in elementary science curricula, especially in under-resourced urban schools [9,10,24]. Indeed,

examples for achieving the ambitious goal of the successful teaching and learning of the NGSS on any large scale are both rare and immensely challenging to implement [25]. There is a clear need to focus educational research and innovation efforts at the system-level in order to develop the effective and long-standing adoption of curricula that can support data literacy skills and model-based learning for all young learners [26].

*1.2. System-Level Considerations for Establishing Elementary Science Reform*

There are several known challenges to enacting broad-scale science reform. One challenge involves negotiating the tension between the need for the widespread adoption of innovative curricula, while also considering the needs of diverse populations and differing local policy contexts across classrooms and schools [1,12,25]. For this reason, systemic reform efforts aimed at the district level, where policy decisions and educational resources within the certain geographic and/or demographic regions are managed, are well suited for addressing these opposing needs [27]. Indeed, research supports the notion that local personnel, such as administrators and teachers, are crucial in determining the capacity for reform [28].

A second challenge to broad-scale reform in elementary-level science education is that reform efforts often clash with school culture in, for example, the low priority (in terms of time and resources) placed on elementary science due to the high-pressure accountability measures associated with math and language arts [12,13]. Furthermore, it is well known that the demands of innovative science reforms, especially those that involve the use of technological tools in support of model-based learning, are especially high for elementary teachers [12,29]. Teachers need high-quality professional development (PD) in order to become more knowledgable in the science content, more familiar with the technology, and more comfortable in the inquiry-based pedagogical approaches needed to successfully implement the curricula [3,11,12,30]. Beyond PD, it is also crucial to continue to support teachers' learning while they are enacting integrated STEM curricula in their classrooms [24]. Finally, research has also highlighted the importance of fostering teachers' social capital, or the capacities that teachers develop through social networks, as critical to the success of reform efforts [13,31].

Given this context, it is important to understand the system variables within a school district that work to either support or impede the enactment of reforms in various settings [1,24,32]. Studying a range of cases and how the capacity for reform changes over time and place can inform the success of future efforts [28,33]. In other words, establishing a research base that clarifies challenges in implementing reform across a range of contexts can equip others involved in science education reform with an understanding of how similar innovations could be adapted in their district, thereby addressing the need to generate more knowledge about "what works, for whom, and under what conditions?" [3]. This type of conceptual approach, known as design-based implementation research (DBIR), includes the following elements: (1) the formation of partnerships among stakeholders focused on implementing innovative curricula that address the persistent problems of practice; (2) the iteration in the design and implementation of such curriculum that integrates diverse perspectives; (3) a goal of understanding the mechanisms affecting the successful enactment of the curriculum; (4) a concern with promoting sustainable change at the system level [33].

Similar to DBIR, the conceptual framework of improvement science also focuses on enacting successful educational reform at scale while underscoring the importance of clarifying the local conditions that produce or hinder improvement [1,34]. The key guiding principles of this framework that pertain to this study include: (1) making a reform effort user-centered, which involves engaging all players involved (e.g., researchers, teachers, administrators) as a network of collaborators from the start of implementation, and (2) a focus on understanding the processes at work within the school system that are in need of improvement [1]. Attending to these principles informs how we can better organize our

educational systems in order to more-consistently improve STEM education for all young learners.

*1.3. Model-Based Inquiry Reform through the Usability Cube Framework*

In this study we employed the DBIR and aspects of the improvement science frameworks to address the need to bring widespread innovative and effective STEM education to elementary students [23]. In accordance with the principles of the two frameworks, this program enlisted a working partnership between a school system, curriculum developers, and education researchers with the goal of co-developing sustainable action plans and curricular units to promote students' understanding of data literacy topics and model-based learning. We conducted a system-level analysis to examine this case, where the participation and engagement of all fourth- and fifth-grade students and teachers in an urban school district were targeted for reform.

One overarching aim of the program was to explore how this integrated STEM curricular approach advanced elementary students' understanding of the epistemic practices of scientists through its emphasis on model-based inquiry and data literacy [35,36]. In this study, we focus on understanding the system processes by which this program was either supported or hindered in terms of achieving broad-scale, lasting adoption by the school district. Importantly, this sustainable districtwide adoption is contingent upon teachers' successful implementation of the program in their individual classrooms. We define the school system as representative of each of the key players in this particular reform effort. The key players in this context include (1) the local ecology center (curriculum developers), (2) the research institute (model/technology developers), (3) the school district (district and building administrators), (4) the teachers and, most importantly, (5) the students.

In order to help clarify approaches in fostering, sustaining, and scaling important, technology-rich reform efforts, such as this one, we employed a usability cube framework as a tool to evaluate its design features and overall characteristics [14]. Similar to DBIR and improvement science, this framework is focused on investigating the conditions under which reform efforts are able to succeed in particular contexts. The usability cube was developed specifically to investigate the workings of school systems engaging in inquiry learning with technology (such as with model-based inquiry) and, therefore, we found it useful as a tool for evaluation in this study [14]. The usability cube consists of a theoretical three-dimensional space that includes capability, school culture and policy/management (Figure 1). Capability refers to the degree that the users of the innovation have, or are supported in attaining, the conceptual and practical knowledge to implement a reform [12]. Gaps relating to the capability dimension can be reduced through lowering the barriers to enactment in the classroom (e.g., in providing high-quality PD to teachers). School culture refers to the degree that a reform effort is consistent with school norms, expectations, and routines [12]. The policy/management dimension of the usability cube refers to the degree that the policies of the district and the management systems that enact those policies contribute to the success of the reform [12].

These three dimensions should be attended to in order to understand the usability of innovations [14]. Each dimension can be plotted on a separate axis, and "an innovation can be placed in this space, where the 'distance' between the innovation and the origin represents the gap that exists between the capacity required to successfully use the innovation and the current capacity of the district" [14]. In other words, an innovation has achieved maximum usability—and thus can be considered a successful reform effort—when system-level players can work to "close the gaps" that might exist. Given this system-level framing, we explore the following research questions with this case study: What were the program design features that either supported (i.e., closed usability gaps) or hindered (i.e., widened usability gaps) the sustainable district-wide implementation of a model-based inquiry curriculum for elementary students? What were the affordances to students who participated in the program?

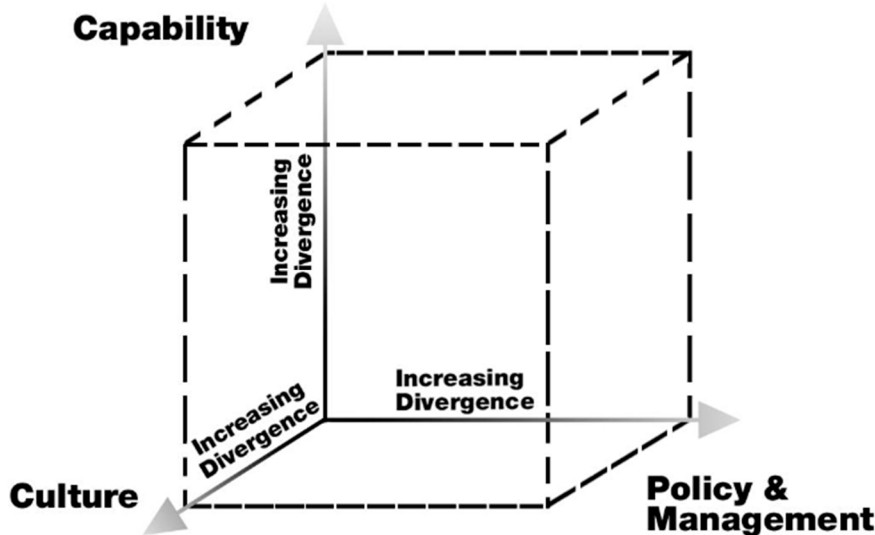

**Figure 1.** The usability cube from [12] (p. 52) and [14] (p. 153).

## 2. Methodology

This is a case study that takes an early stage, or exploratory, noncomparative qualitative approach to addressing the research questions, where we examine approaches to problems in education to inform the basis for the design and development of new learning interventions [37]. We focused our investigation on elucidating the program design features that either supported or hindered the sustainable districtwide implementation of a model-based inquiry curriculum alongside evidence that points to student affordances due to program participation. This study was part of a larger research program, conducted from 2017 to 2020, that aimed to develop curriculum and instruction activities to support learning about complex systems with elementary school students (Institutional Review Board approved protocol #831918).

### 2.1. Context

As part of a collaborative venture with a large research institute in the Northeast and a small ecology center located in the Midwest, the partner entities worked to bring curriculum grounded in model-based inquiry, innovative technology integration, and data literacy topics to all elementary students (4th and 5th graders) in a large, urban midwestern school district. The research institute and the ecology center collaborated to develop six model-based inquiry modules (one introductory module, two modules for the fourth grade, and three for the fifth grade) linked to the science standards mandated by the district as well as the NGSS. Topics of these modules include a general introduction to complex systems and how they can be modeled to simulate real events (such as the spread of an infection), how to examine survivorship patterns and the relationships of various organisms in particular ecosystems, and how to model ways to mitigate or control the amount of pollution in the environment (see Appendix A for links to the full curriculum lesson plans). The curriculum was built around the use of the StarLogo Nova simulation platform, which is a graphical programming software that enables students to learn simulated behaviors of individual agents and to observe emergent properties of a system as agents interact with each other and the environment [38,39]. StarLogo Nova affords students the opportunity to run multiple simulation experiments to collect and analyze data and thus represents a dynamic, visual learning technology featuring system aspects and interactions that would not be available through typical learning modes (e.g., static images and text-based descriptions) [22].

To support systems modeling learning and inquiry as an instructional strategy, participating teachers within this school district participated in extensive PD to deepen their understanding of science content, to build comfort with the StarLogo Nova modeling

environment, and to foster pedagogical shifts to facilitate instruction. Addressing these features through carefully designed PD was important, since it is known that teachers often exhibit low confidence in teaching science content without formal training and can also find it difficult to teach new, innovative pedagogical approaches without practical advice and exposure in how to teach it [40–42]. Thus, the PD cycle for a given cohort of teachers included two weeklong 30 h workshops—one held in the summer before initial implementation (introductory workshop), and another the summer afterward (advanced workshop). Each workshop included a mix of hands-on experience using the StarLogo Nova modeling tools, structured opportunities for teachers to provide usability feedback on the curriculum modules, curriculum planning time, and engagement with various ecology center staff and researchers. The advanced workshop supported teachers in consolidating and reflecting on their learning over the preceding year and was complemented by structured opportunities to plan for the year ahead. In addition, after-school sessions (totaling 12 h) were interspersed throughout the academic year and provided additional support for the implementation of each module and to develop advanced modeling and classroom leadership skills. Finally, in addition to developing all of the training protocols and processes, described above, the program developer and the program coordinator (both employed at the ecology center) worked in participating teachers' classrooms during the academic year in a variety of capacities. Given that teachers had differing comfort levels enacting the modules, the scope of the program staff's involvement ranged from modeling lessons in order to demonstrate the pedagogical strategies needed (for less confident teachers), to simply being a source of ancillary support during investigations led by the teacher (for more confident teachers). In short, this PD included aspects such as (1) modeling effective instruction, (2) offering coaching and expert support, (3) dedicating time for feedback and reflection on practice and (4) sustained duration of PD participation during the academic year. These aspects are known to be crucial in engaging teachers in effective and high-quality PD [43–45].

### 2.2. Participants

Over the course of the 3-year grant funded program, a total of 37 teachers across six schools (representing all the elementary schools in the district) were trained in the model-based inquiry program. Since one goal of this study was to understand the design features of the program that supported or hindered teachers' sustained implementation of the StarLogo Nova curriculum, we interviewed and conducted classroom video observations of six fifth-grade teachers who volunteered to participate in the study. While there were no fourth-grade teacher participants in this study, the six fifth-grade teachers represented a broad range of teaching experience (from approximately 1 to 10 years of experience). This allowed us to sample various perspectives concerning implementing model-based inquiry in the classrooms, as teachers with more experience have been found to be more proficient than novices in accomplishing teaching goals and in enacting high-quality instruction [46,47]. As for the student population, this district served a diverse urban community, where 99% of the students attending elementary schools received free or reduced-price lunch (more demographics are provided in Table 1). Therefore, the sociodemographic composition of this district illustrates that this reform effort was targeted towards a population of students that is in critical need of improved educational resources in STEM subjects [1,5,10,24]. In order to better understand the affordances to students who participated in the program, we observed and video recorded two lessons from the six classrooms of the teacher participants. In total, there were 114 fifth-grade students spread out across these six classrooms, and we selected four students from each class, working in pairs to create a sample population of 24 students (13 females, 11 males) to analyze their learning experience as they worked with the StarLogo Nova simulations and collaborated with one another. Of these 24 students, ten identified as White (non-Hispanic), six as African American (non-Hispanic), three as Hispanic, and five as multiple ethnicities. These students were selected in conjunction with the classroom teacher for characteristics

that would enable us to capture high levels of dialogue and interaction during the activity. In addition, we interviewed the program developer (Rich) and the program coordinator (Sarah), who both worked at the ecology center. Rich and Sarah provided rich insight into the system-level relationships between key stakeholders in this investigation (i.e., the research institute that developed the models, the participating teachers, and the district and building administrators). Names used to identify participants in this study are pseudonyms.

**Table 1.** District demographics within each school.

| School | School Enrollment | Black | White | Hispanic | Two or More Races | Asian | American Indian or Pacific Islander |
|---|---|---|---|---|---|---|---|
| School A | 537 | 42.8% | 27.7% | 18.2% | 9.5% | 1.3% | 0.4% |
| School B | 504 | 39.5% | 31.5% | 16.7% | 11.5% | 0.6% | 0.2% |
| School C | 500 | 36.8% | 23.4% | 27.4% | 11.4% | 1% | — |
| School D | 515 | 48.5% | 17.3% | 22.5% | 10.3% | 0.8% | 0.6% |
| School E | 363 | 29.2% | 55.9% | 12.1% | 2.8% | — | — |
| School F | 494 | 36.8% | 30.2% | 21.5% | 8.1% | 3.2% | 0.2% |

*2.3. Data Sources and Analysis*

We conducted individual, semi-structured interviews with the six teachers, the program developer (Rich), and the program coordinator (Sarah). The eight interviews lasted between 45 and 60 min each, were recorded and transcribed for analysis, and resulted in approximately 7 h of interview data. Interview questions for teachers aimed to probe for the challenges and affordances of enacting model-based inquiry curricula in the classrooms. For example, teachers were asked questions such as (1) What contextual variables do you think might be barriers to implementing this project for you, or for other teachers? (2) Do you feel that students can gain the same knowledge and experiences using paper and pencil, or other tools, as they do from using the StarLogo Nova models? Why or why not? The interview questions for Rich and Sarah probed for similar concepts, and also sought to provide deeper insight into the larger system variables (i.e., variables beyond the classroom) that supported or impeded their goals for the program. For example, they were asked questions such as (1) What was the nature of the partnership between the ecology center, the research institute, and the school district? In retrospect, what could have improved the partnership and helped sustain the program? (2) What would you describe as the affordances of bringing model-based inquiry to elementary science teachers? Were the affordances achieved?

The study questions, framework, and data-collection techniques were constructed so as to maintain as naturalistic a methodology as possible [48]. The interview data went through a systematic process of validation. First, we used a modified grounded theory approach in evaluating this data source, where we searched for evidence of the a priori categories related to student affordances and the usability cube dimensions (i.e., capability, school culture, and policy/management) as either positive (i.e., decreased the usability gap) or negative features of the system (i.e., increased the usability gap) [49]. Using a constant comparative method of analysis, the first author read the first interview where information that related to each of those categories was derived, and that information was then compared to and triangulated with each subsequent interview in order to validate that a particular finding emerged from multiple sources [49]. We also triangulated the evidence that emerged across the separate perspectives of the teachers, Rich, and Sarah. Gathering interview data from both the participant teachers and the program coordinators represented emic (or within social group) and etic (or outside social group) perspectives, which allowed for a more holistic picture of the supports and challenges that teachers encountered in their program participation [50]. The second systematic way we validated the findings was to work with authors 2 and 4 in order to negotiate and resolve any discrepancies in

categorization [51]. Third, we used member checking in which we validated the findings with two of the interviewees that were available, and this helped to ensure accuracy and validity in our representation of the findings [50].

Where possible, we also corroborated findings that emerged from these interviews with evidence from video recordings of some students participating in the model-based inquiry lessons. The twelve pairs of fifth-grade students (two pairs per each teacher interviewed for this study) who were recorded each engaged in two modeling lessons, for a total of ~20 h of videos of students engaging in model-based inquiry; this footage is used as the primary data source in another study (under review) on students' epistemic performance in developing data literacy skills. Here we draw on this data source largely to triangulate themes that emerged from interview data with respect to research question 2: What were the affordances to students who participated in the program?

## 3. Results

Here we present the major themes that emerged from the emic perspectives of the six participating teachers (the users of the innovation) and the etic perspectives of program coordinators (the curriculum and PD developers). We also triangulate these findings with observations from classroom video recordings when possible. We employed a usability cube framework as a tool to understand the variables effecting the capacity for the sustainable implementation of the model-based inquiry program. We found that there were multiple features built into the program design that decreased gaps within the three usability cube dimensions and thus promoted the short-term success of the program. These features were a strong partnership among the program developers and the district (capability), strong teacher support networks within some schools (school culture), and districtwide communication about and support for the program (policy and management). However, other events surrounding the program also served to increase gaps, and these were consequential enough to have contributed to the lack of widespread and sustained adoption of this reform effort within the school district. These included a lack of teacher comfort with the new curriculum (capability), limited peer support and time (school culture), and high teacher and administrative turnover (policy and management). Despite not being able to achieve sustainability beyond the 3 years of funding awarded to the program developers, there were notable positive outcomes associated with student learning (e.g., participating in innovative and exciting science learning environments and practicing data literacy skills).

### 3.1. Capability

The main factors that emerged to decrease the capability gap, and thus contribute to the temporary success of the program, were (1) an active working relationship and partnership among the program developers, research institute and the district, (2) a program curriculum that was well-aligned with relevant standards, and (3) easy-to-access and well-organized curricular resources.

In this program, the strong collaborative partnership between the research institution that developed the technology and the school district that implemented it was facilitated by the local ecology center, where both Rich and Sarah worked. In articulating the overall role of the ecology center, Rich said, "So we act as a broker, if you will. We're essentially bringing [cutting-edge] technology to a midwestern school district and making some change in there. The school district benefits, they get access to things that they wouldn't otherwise have." Sarah elaborated on her experience as project coordinator, "[The research institute] was really terrific to work with. I worked closely with the person who built the [StarLogo Nova] models and we emailed back and forth, we'd make phone calls. It worked out really well . . . . So [the research institute] didn't have anything to do with the district really. I was really a liaison kind of between all the people. However . . . we had a good working relationship, and I felt like I had the support of everybody who worked on the project."

In fact, Sarah had over 20 years of teaching experience (including 6 years of elementary-level teaching experience in the district) before she became project coordinator, and she was already well-known by many teachers and district administrators. She also served as an ad hoc member to the district faculty committee tasked with revising the elementary science curriculum and had been critical in helping teachers articulate a need to "institutionalize" inquiry-based modeling. Therefore, while the research institute had the expertise to develop the models themselves, the local ecology center provided feedback on the new models being developed; this feedback helped the research institute understand user constraints and conceptualize better ways to use data visualizations to complement the agent-based representations in their program's learning context (particularly with regard to younger students and the limited STEM background of many elementary school teachers). The ecology center also provided extensive user testing both in the context of the teacher workshops as well as in game design camps for kids, which the ecology center ran in the beginning stages of the program. This collaboration between the curriculum developers and the users to improve the design of the learning environment starting from the earliest stages of implementation is an essential feature of both DBIR and improvement science [1,33].

It was through this collaboration, and with significant input from district leaders, that Rich and Sarah were able to work with the research institute to ensure the curriculum was a good fit with the district needs and interests (e.g., direct links to district-identified topic areas and the mandated science curriculum standards that teachers were accountable for) as well as with NGSS standards. In accordance with DBIR and improvement science approaches, the curriculum modules were then continuously revised based on teacher and student feedback to ensure that the materials were better calibrated to district, teacher, and student capacities. Teachers found these links to the district standards to be a valuable factor in their ability to implement. For example, Teacher 1 stated, "I don't feel like I have to stop teaching science because I have to get [the modeling curriculum] in. I really do think it ties in together." Teacher 2 confirmed that "I think the connection piece to the science curriculum with [the district] is really amazing, and I think it would benefit the students to do this all year round, not just with one standard." Teacher 3 added that:

> It really encompasses all the skills and concepts that [the students are] learning, and they get to put it to use . . . The visual model, it helps them take the concepts and the vocabulary words and everything that we're learning and putting it to use and seeing how it is actually used in real life . . . This really amped up [student] knowledge and really I think it checked off all of the markers of the standards we are teaching right now.

In addition, each of the new modules provided to the teachers included access to the model environment in StarLogo Nova, as well as extensive curriculum support (including links to pertinent standards, recommended instructional sequences, possibilities for extensions, embedded assessments, and links to related curriculum resources). The embedded content and pedagogical supports were comprehensive and easy for the teachers to refer to during implementation. For example, Teacher 4 stated:

> Everything was pretty much laid out for me. And then you have everything in the Google drive shared with us . . . I don't have to go searching for things and piece it all together . . . after the summer [PD] you kind of forget everything. And once I went back through the binder and then went back through the lesson plans, it just all comes back and everything is laid out exactly what to do. So that was really helpful.

Teacher 6 added, "I really think that how you guys laid out the lesson plans, that was amazing . . . when it's already built in as lessons, then it's pretty much there for us [to use]." Developing these easy-to-access, standard lesson guides was likely key to reducing the stress and cognitive load that accompanies new and complex tasks [1]. Therefore, the clear alignment with standards, well-developed lesson plans, and easy-to-navigate resources

provided teachers with some of the required capability needed to enact the model-based curriculum in their classrooms.

Despite these efforts to close the capability gap, two factors served to hinder teachers' capability in using the curriculum. The first concerns the multifaceted STEM competencies needed to teach model-based inquiry effectively. The second, and related, factor concerns teachers' lack of confidence or comfort in implementing the curriculum independently (i.e., without the help of Sarah coming into the classrooms to lead the lessons with students).

The STEM competencies needed for optimal instruction within this program required familiarity with the underlying science concepts being modeled; fluency with data representations, technology, and statistics; and classroom leadership skills to facilitate evidence-based discussions that emerge from small-group discussions. Teachers' conceptual and practical knowledge in all of these areas influenced the success of the model-based inquiry venture. However, according to Rich, a typically trained elementary-level teacher is not well prepared for the multilevel STEM competencies needed to teach effectively with models. Most elementary-level teachers go through a very general preparation program and receive certification without a specific disciplinary focus. This lack of specialization in STEM areas is compounded as teachers progress through their careers, as they need to remain current in best practices across a range of subject areas. Rich noted that they were "asking teachers to change their pedagogy, their comfort with technology, their content understanding all at once. That's a lot of change to ask a teacher to do . . . but [this] specific challenge that I found over the years with elementary teachers, is because they're not specialists in [a particular discipline]."

Indeed, this lack of depth in science teaching training appeared to negatively affect teachers' confidence and comfort in implementing the lessons. Sarah indicated that she "did the majority of the teaching . . . especially in the beginning for teachers. They wanted to feel more comfortable with it and that seemed to be the way to raise their comfort level with the curriculum and the models." In fact, 78% of the participating teachers (29 out of 37) never ended up teaching the curriculum on their own, instead relying on Sarah to implement the lessons in their classrooms. Evidence of Sarah's classroom leadership role was supported by the classroom video recordings where we observed her leading in each of the observed lessons. Her teaching involved a preamble to the lesson (~20 min) in which she explained the learning objectives, elicited students' prior knowledge, and demonstrated the modeling activity, followed by student pairs working independently with the models (~30 min), and finally a conclusion with a whole-class discussion to solicit students' understanding of data literacy (~5 to 10 min). Teacher 5 noted that she was "really uncomfortable with the idea of [teaching the lesson herself]" and that even when she got to the point of teaching it herself that "it wasn't great because [she] didn't really know what [she] was doing." She further reiterated that this curriculum is not for every elementary-level teacher and that without a solid "technological background" it would not be navigable. Sarah confirmed Teacher 5's comment, stating that "one of the greatest fears out of most of the teachers, was even though they'd had lots of instruction on how to use the models, they just couldn't make themselves take that next step to actually teaching the lesson because they were scared they couldn't answer the questions." Despite the discomfort that the majority of teachers experienced with the modeling aspect of the curriculum, a few teachers (8 of 37) took greater ownership and exceled in their implementation, but it turns out that a particular feature of their school culture was apparently key to their success, as we discuss next.

### 3.2. School Culture

Our findings revealed one crucial factor related to school culture that decreased the usability gap and thus contributed to the autonomy that teachers needed to enact the curriculum in their classrooms. Specifically, in two of the schools, six of 37 participating teachers were placed into in-school teacher teams of three by their school administrators. Even though these teams were created by the school and not tied specifically to the program, they allowed space for these teachers to readily collaborate with and support one another

in implementing their lessons. Sarah noted that "all the [teachers] that were teaching it on their own, except for a couple . . . was because they all decided to do it all together and they had each other to support . . . teachers were much more likely to stay with the curriculum if [peers within their school] were participating." This is in stark contrast to the teachers who were *not* part of in-school teacher teams and who, with the exception of two cases, always needed sustained, in-class support from Sarah. Teacher 3, who eventually taught the curriculum on her own and was part of one of the in-school teams, corroborated this point, commenting that "it's helpful when it's . . . your whole team doing it because then we can really talk about that. I think it would be a lot harder if it was just me on my team doing it."

Two factors related to school culture that increased the usability gap were the difficulty in creating a strong peer network across teachers at different schools and not having enough time for science in the school day. Rich and Sarah made deliberate efforts to cultivate a peer network for cohorts, but this venture was largely unsuccessful. For example, they initiated a peer mentor program with four distinguished teachers from the first cohort of teachers (i.e., those who participated in year 1 of the grant). These teachers were comfortable in enacting model-based inquiry in their classrooms and had the "political savvy" needed for working with building and district administrators. They participated in the summer workshops for beginning teachers and were available to peers as they began or continued their work with the curriculum. According to Sarah:

> We were hoping that [these four teachers] were going to be the ones that were going to help this continue once I had to pull out once the grant was over. And it just never took off . . . we talked with the administration about . . . pay[ing] these teachers a little bit of a stipend so that they could be ambassadors and they could go to the other school buildings and help train other teachers who were interested, but it just never happened. I think that would have made a big difference too. That could have helped it take off.

In addition, Rich and Sarah tried to bring participating teachers together in other ways, such as through short (approximately 2 h) meetups dispersed across the school year. The idea was for teachers to share their teaching experiences and challenges and build a stronger peer network with their cohort members. However, Sarah noted that regular school year meet-ups did not really work because "teachers are just swamped . . . trying to carve out a little bit of time to ask them to do something extra is really hard to do . . . Some of them didn't feel like they could give the program their all, even though they really wanted to."

Time was also a constraint within the actual school day for teachers. Allocating adequate time to teaching science was another factor related to school culture that hindered the success of the program. Science was often assigned to short (~25 min) time slots near the end of the school day. Teachers needed more time to instruct students in the deeper learning required for the model-based inquiry lessons. Teacher 6 stated that "time is always an issue . . . we haven't really jumped into science too much. They push reading and math pretty hardcore . . . Most teachers don't agree with it, but we're kind of at the mercy of what we're told." Teacher 5 had a similar experience in her school, noting that "it was hard to fit it in, in the day . . . Because it was either, social studies or science and we have to hit social studies every day . . . And so then, it's like well I don't know where science goes." Teacher 2 confirmed that "the scheduling was just difficult because we didn't really touch base with scientific things until we did [this curriculum] . . . I think if we had a set schedule where science was implemented throughout the Fall it would've been easier." The struggle to fit science in during the school day detracted from teachers' ability to reliably implement the curriculum at the depth needed for meaningful learning.

*3.3. District-Level Policy and Management*

Rich and Sarah invested considerable effort at the district-level to decrease the usability gap by communicating regularly with administrators to gain their support for the

program. This is known to be a critical step towards helping to ensure sustained, districtwide implementation of the program [28]. Rich indicated that, overall, district-level administrators supported the idea of the program, and he said that their support was facilitated by Sarah's familiarity with the district. In fact, Sarah noted:

> I knew the superintendent really well because he was my principal when I first started in the district . . . And I had a good relationship with the people who were in charge of the curriculum as well . . . the head of curriculum, he really tried to get all of his principals on board with this.

In her words, administrators were overall "accepting of the program." In the first year of the project, Rich had "constructive dialogue" with the assistant superintendent, who demonstrated support by engaging principals and marshalling needed resources. Before the start of the second year, Rich and Sarah met with all the elementary building principals, the assistant superintendent, and the district science coordinator and, at that meeting, gave an overview (~1 h) of the program so that everyone was familiar with it and any concerns about teacher and student assessment could be addressed. They also sought feedback from the principals in terms of guiding the program design. In addition, they met with the district's Board of Education (~30 min) to inform them about the program and discuss details regarding teacher training.

Despite these efforts to solicit administrative buy-in, one critical district-level factor emerged in our findings that likely substantially increased the usability gap: high teacher and administrator shuffling and turnover in the district. In particular, teacher shuffling involved frequent grade-level changes imposed on teachers, which contributed to a lack of cohesiveness in the peer network while also decreasing teachers' incentive to participate. In other words, this habit of grade shuffling made it difficult for teachers to support each other in undertaking complex reform while operating in a continuing state of flux. Of the 37 teachers who had started in the program, 12 (32%) dropped out due to involuntarily grade-level changes imposed by their principals. Sarah reiterated this concern, saying that that in one problematic year "we lost all of our fourth-grade teachers and two of our fifth-grade teachers." Rich commented that "the district has an interesting pattern of constantly cycling the teachers into different grade levels, and so that works against [sustainable reform] . . . [Teachers] know [they're] going to be out of this grade in a year or two anyways, do [they even] invest in [the program]?" Adding to this state of flux, in the third year of the program seven of 12 school administrative positions (i.e., principals and assistant principals) were filled either by people in their first year in the role or by people who had switched to a new school. As a result, most of the new school administrators were either unfamiliar with the program and its expected role in the district, or they had different goals in their new position and the model-based inquiry program was not a part of it. Rich commented that "support varied widely among [these administrators]." Sarah agreed that stronger support from these administrators would have likely helped the program become more sustainable. She noted that "administrative support from the principals, if we could have gotten more buy-in from the principals, that definitely would have made a difference."

Both Sarah and Rich lamented that, in bringing model-based inquiry to elementary students in the district, the sustainable program they initially envisioned never took off. Despite their efforts to recruit teachers and foster district-level support, they could not get every 4th- and 5th-grade teacher in the district to participate, which was their ultimate goal. Sarah observed, "I felt like we did a really good job working with the students and the teachers who we had access to, but it was based on their interest and whether they wanted to participate or not." The teachers also felt that the work they were doing was important and should be more widely implemented. For example, Teacher 1 said, "I think that [this curriculum] is what this generation is going to be doing in the future . . . A lot of them are going to have to do lessons online in the future . . . so I do think that this helps prepare our kids for the future better than the curriculum I have now." Teacher 2 added, "This program is really cool. I wish you guys could get more funding and more grants and

expand on this. I think this would be fabulous to do for ... most of all the grades in some way, shape, or form." However, as Rich stated, "Once the funding ended, the program ended, and while there might be one or two teachers that have carried [the curriculum] forward ... in any meaningful form it's not there ... There's nothing in the district that encouraged [sustainability]."

*3.4. Affordances to Students*

Notwithstanding the cessation of the program after the 3-year funding cycle, there were noteworthy affordances to the learning of students who engaged in it. Sarah and participating teachers were able to create technology-rich, active learning environments for their students. Sarah commented:

> Teachers ... realized, 'this is really cool, students are helping each other ... ' [so] the students were doing some of the teaching as well. They were working together ... And pairs would work together, but then you'd hear somebody over in the corner asking a question, this kid would yell across to her, 'Hey, well do this.' It was great because the teachers weren't saying, 'Shh. Shh. Get quiet.' They just let them engage.

Teacher 2 also emphasized the active learning benefits of the curriculum. She added:

> I really like to see [the students] cooperate with each other, that is something that, it's definitely hard in this classroom, especially because there's a lot of kids that have difficulty with that. However because they're very interested in the StarLogo [Nova] ... modeling, the manipulation of it all. They share really well ... And there were some students who don't normally do well with partner work, but they did really well when we did the partner work with the [modeling curriculum].

The classroom video recordings further supported this finding, in that as soon as Sarah discussed the learning objectives and demonstrated the activity with the whole class, students began working in pairs and having lively and epistemically rich conversations with one another. We observed that 10 out of 12 pairs of students engaged in dialogue where they collaboratively developed and justified reasonable predictions and/or made accurate inferences regarding the garden ecosystem they were experimenting with in the model [35]. For example, Student 2, discussing with their partner how to predict the number of surviving turquoise plants in their next model run, said, "I think [the] turquoise [plant is] going to get up to like 220." Their partner, Student 1, responded, "I think 150," and Student 2 countered, "Are you sure? That means it would start dying ... so wanna go for 205?" Student 1 nodded, and they both wrote this prediction on their datasheet (29 October 2018). In this exchange, students actively negotiated a reasonable prediction (i.e., the epistemic aim) based on evidence.

Furthermore, for elementary students in this district to be able to manipulate the variables within the modeling curriculum represented an important educational opportunity that these students would otherwise not have had. This theme was highlighted throughout all six of the teacher interviews. Teacher 1 remarked that "it gives a chance for the kids to manipulate it at their own pace. It gets them [an opportunity] to see something they wouldn't normally see, like the plants growing. And I think it really reaches out to different kinds of kids." Teacher 2 observed that "it's more than just a picture or a video, it's a model they can manipulate as well. And I think that when kids are able to manipulate things it kind of solidifies it in their brain a little bit more, if they're able to play and if they're able to switch things around." Teacher 4 echoed this observation, noting that "being able to see and manipulate that model was pretty priceless." Teacher 5 said:

> Modeling helps with their hands-on approach ... They see in the instant ... as they change something, something happens immediately they have that immediate reinforcement or negative of changing something and it not working anymore

. . . and being able to tinker with it . . . being able to see immediately what their choices have done, what their impact is [helps students learning].

We also observed instances of students in the classroom videos exploring new questions related to the model on their own, beyond what they were required to investigate for the class activity. For example, a pair of students who had finished running their experiments on the garden ecosystem model, where they varied the rainfall but kept the amount of sunlight constant (at full sun), decided that they wanted to tinker with the model variables further. They changed the sunlight "to see what [would] happen" in the garden when they set the rainfall level to 0 (or drought) and the sunlight level to 0 (or full shade). When they ran the model and observed the outcome one student said, "That's magical!" and the other responded, "So [the] yellow [plant] is growing, and red! I didn't know [the] red [plant] grew!" The students were surprised to discover conditions that allowed the red plant to grow well, because in their previous experiments the red plant had not survived in large numbers compared to other plants in the garden ecosystem (26 October 2018). Thus, teachers noticed that their students were able to productively collaborate with one another, actively change variables in the computer model, and then were able to see the effects of those changes immediately. These affordances were also captured on the video recordings. Teachers agreed that this model-based inquiry instructional approach was not only important in promoting student learning, but also represented a new and exciting learning environment that they had not likely experienced before.

Teachers also confirmed that, beyond being able to manipulate models in real time, other affordances, such as the opportunity to build on data literacy skills, were unique to this curriculum. Sarah noted that students "learned about data and how to collect data and how to use it, how to read it, understand it . . . students don't really have the opportunity to [do this]." Teacher 6 confirmed that the students "learned a lot about data and data collection, and why it's important . . . I think they even became a little bit more independent with looking up data." In addition to those kinds of contextual experiences (i.e., in understanding how to collect data and where it can come from), students were also able to practice interpreting various data visualizations. For example, Teacher 1 stated that "they were really just, naturally, finding patterns amongst their data without realizing it . . . without even realizing that they're analyzing data, they are . . . they [also] looked at the graphs really heavily. They liked looking at [them] and seeing an increase or decrease." We noticed this behavior in classroom videos as well, where students would point to the dynamic graphs on the screen and describe the plant survivorship trends over a particular growing season. Students made comments about growth patterns "seesawing" (26 October 2018), "skyrocketing" (25 October 2018), and "staying the same" (25 October 2018).

As mentioned briefly above regarding student collaboration, teachers also noticed their students using evidence-based reasoning to make accurate inferences and reasonable predictions. Teacher 3 stated that students "had to collect evidence and they made inferences based on that evidence, [for example], which plants represented which color in the model," and Teacher 5 noted that her students "were making a lot of inferences" throughout the lessons. Teacher 4 observed that the modeling helped students to reason through data, "especially with making predictions . . . running the model multiple times and then [having] to make the predictions. And then kids would share. And I think that [getting] input from other kids . . . that definitely helped them [practice this skill]." Video data corroborated these findings, as described above with 10 of 12 student pairs engaging in epistemically driven negotiations. As an example of students making accurate inferences, we observed one student saying that "[the] yellow [plant] must be a succulent because it doesn't like that much water" (30 October 2018). In fact, all 12 pairs of students were observed using the data they collected from the model to make accurate inferences about plant ecology [35]. Therefore, based on teachers' perspectives as well as classroom video data, it is clear that the model-based inquiry program allowed teachers to enact instruction that promoted data literacy skills with their students.

Finally, all six teachers mentioned that the engagement level was high across their students during the lessons, and that the curriculum was able to bring out the best in a wide range of students. Sarah stated that "students who normally struggle in regular, typical classroom scenarios, some of them just flourished with the [program] . . . so I think there are lots of really positive experiences that came out of it for all different levels of students . . . [when] they ran those models, you could see their interest get sparked." Teacher 1 elaborated that she "noticed some kids who don't usually get excited to participate in science, really wanted to do this project. I have one [student] and it's really hard to keep him in class but he gets pretty excited when he sees [Sarah] come in. So I think, really, the engagement piece is pretty big." Teacher 6 added that "even kids who normally are not engaged in lessons, and are working on a lower level, they can make connections to this . . . I really just think that it can reach many students. Not just a particular group, but I've watched it reach all levels of students in my classroom." Teacher 5 confirmed that "[the curriculum] really gets the kids interested and it can reinforce the things that they're learning . . . I enjoyed it, my kids really enjoyed it, I think they're getting a lot out of it." The classroom recordings supported these statements, as all 12 pairs demonstrated interest and curiosity while interacting with the model. For example, students would often celebrate when they made successful predictions. In one instance a student yelled to their partner, "Oh, we were so close to turquoise!" and their partner excitedly replied, "We chose 205, and it came to 209!" Then one of the students called the teacher over, and both proudly showed their work and explained how close their predictions were to the model output (25 October 2018).

This program overall was successful in engaging students in inquiry, active learning, and data literacy skills. However, whether the positive student learning outcomes described above led to an increase in student interest in scientific inquiry or complex systems modeling was not assessed. As Rich put it, "that's another challenge with [this type of 3-year project], that [we] don't have the [ability] to really track the long-term change." And while he recognized that these short-term affordances to students occurred, he wished that he and Sarah "had a bigger footprint, where [this program] became the entry point to more extensive use of [model-based inquiry]."

## 4. Discussion

In this study we applied a usability cube framework to evaluate the program features that either supported or hindered the districtwide, sustainable implementation of a model-based scientific inquiry program. In accordance with a DBIR and improvement science framework, we addressed a persistent problem of practice (i.e., the need for innovative elementary science curricula), which was tested and iterated to better incorporate the needs of the district, teachers, and students, with the goal of clarifying the system processes affecting the long-term adoption of the program.

### 4.1. Features that Increased Program Usability

We begin with a discussion of the prominent features that increased the usability of the program. These features included a strong working partnership among the program developers, technology developers, and the district, and the availability of high-quality PD and teacher resources.

The partnership between the local ecology center, the research institute, and the district was essential in helping build the capacity for curricular innovation [1,33]. In order for curricular innovations to be successful, it is important that such partnerships are in place to ensure all the players can work to create shared goals with concrete plans to achieve them; otherwise, widespread program adoption is most likely to fail [14]. In this context, Rich and Sarah were successful in collaborating with both the research institute and the school system to create strong curricular links to local science standards and district-identified topics (e.g., in the use of local plants in the garden ecosystem model). This alignment with both local standards and localized, familiar topics within the modeling

activities provided teachers with an incentive to participate and also fostered personalized connections to the curricula [23,25]. To integrate local policy and learning context in this way, the partnership involved frequent iterations of curricular design features based on administrative, teacher, and student feedback, and such iterations are known to increase the usability of programs [1,33].

In addition, Rich and Sarah understood that teachers' gaps in content knowledge and pedagogical knowledge, as well as some apprehension about teaching with new technologies, are well-known barriers to elementary teachers' capacity for implementing new curricular innovations, such as those involving model-based inquiry learning [11,12,14,29]. To mitigate these barriers and thus increase teachers' capability for enacting innovation, Rich and Sarah engaged teachers in high-quality, ongoing PD as well as provided classroom support and well-organized resources and lesson guides. The objective of this program feature was to build teacher learning and confidence over time through modeling pedagogical strategies and developing teachers' social capital, so that eventually teachers would take up the curriculum independent of on-site support from the ecology center. Indeed, previous research has demonstrated that the ability of teachers to form and rely on peer support networks is potentially integral to the successful implementation of urban STEM education reform programs [31,52]. Furthermore, strong teacher learning communities have been linked with the increased likelihood that participants will lead instructional reforms within their districts [53]. These capacity-building qualities were evident in this study, where only the teachers who were placed in "teams" (with the exception of two others) developed the confidence and expertise needed to independently enact the model-based inquiry activities. Moreover, two of the teachers from these teams ultimately became ambassadors for the program, who took on facilitator roles in later PD cycles and were slated to help sustain the program after the funding period ended, although this final phase never came to fruition. Because the district leaders neglected to recognize these teachers as valued improvement leaders in STEM education and were unable to remunerate them for working to continue in the program, the goal of attaining a scalable and sustained model-based inquiry program in the district was not realized [1].

This study also demonstrated that there were multiple student affordances associated with the program. Both the teachers and Sarah and Rich agreed that notable strengths in the program were the ability to (1) bring new technologies and inquiry-based, active-learning experiences to students, (2) engage students of all levels, and (3) improve data literacy skills. Engaging students in active learning is known to support diverse learners and can help ameliorate achievement gaps that often exist between minority and/or low socioeconomic status groups and their counterparts, (e.g., [54,55]). The use of computer-based simulations and modeling is known to increase the emotional engagement of students in learning science and result in more inclusive teaching practices [56,57]. In addition, and consistent with other studies employing this StarLogo Nova technology with young learners, these students were able to demonstrate the data interpretation skills that are crucial, yet challenging, to begin cultivating effectively at a young age [58–60]. Therefore, this program provided teachers with high-quality, NGSS-linked science curricula, which are scarcely available in elementary education; furthermore, it helped to promote critical aspects of inclusivity and data literacy in their classrooms [61].

### 4.2. Features that Decreased Program Usability

We uncovered several important impediments that increased usability gaps and affected the capacity of the district to scale and sustain the reform. For most teachers, the challenge of implementing model-based inquiry independently was not easy to overcome, and this is likely explained by their inability to develop the social capital that was a characteristic feature of the successful teachers. Even though Rich and Sarah tried engaging teachers in ongoing PD throughout the academic year in order to build on the peer relationships necessary for success, it was difficult for most teachers to activate their social networks to support capacity building. One explanation for this discrepancy could involve

the varying school pressures that hindered teachers' collaboration, so that teachers at different schools were more restricted in their ability to engage with peers around the goals of the program [62]. For example, teachers often did not have the time to attend schoolyear meetups facilitated by the ecology center. Thus, the less frequent opportunities for those teachers to interact and share program-related resources and insights was likely an important limitation [13,63]. Perhaps working more closely with administrators to develop supportive norms for teachers' participation in such learning communities—norms that serve to increase teacher interaction and communication—would have incentivized more teachers to engage in the peer networking opportunities available to them [13,62,63].

Another prominent barrier to building reform capacity was the insufficient time allocation and prioritization placed on elementary science learning within the district. While allocating sufficient time for deep learning is a necessary component of successful science reform [24,64], science historically has, and continues to be, a fringe subject that is largely sidelined behind math and reading in elementary education (e.g., [13,61,65]). This lack of prioritization was apparent in this study, with the teachers indicating that the accountability pressures they faced were greater for other subjects. To help close this usability gap, future programs aimed at reforming elementary science might consider aligning activities more explicitly with math and literacy standards and reimagine the reform effort as a means to contextualize these higher priority subjects into a more integrative STEM education approach [33,53,61].

A third barrier to program adoption and diffusion was the frequent shuffling and turnover of administrators and teachers through different schools (administrators) and grade levels (teachers) within the district. The involuntary attrition of approximately one third of the teachers from the program represented a significant loss in investments intended to foster teacher learning and development and to promote the broader adoption of the curriculum [66]. The lack of stability among administrative positions further diminished the initial effort put forth in establishing the district support needed to sustain the program. Administrative advocacy plays an important role in setting the stage for reform [26]. Certainly, coordinating efforts and resources at the various system levels and adhering to a shared vision becomes increasingly difficult when the roles of key players tend to shift [12]. Both Sarah and Rich agreed that an increase in support on behalf of the principals might have helped build capacity; however, as is often the case in large urban districts, principals, whether or not they were new to their roles, were likely more concerned with seeing teachers boost standardized test scores than seeing them implement ambitious science reform [33].

Taken together, we found that the lack of widespread activation of social capital combined with low value placed on innovative science curricula and continual turnover of administrators and teachers within the district increased the usability gaps across all three usability dimensions (capability, culture, and policy/management) to a degree that left the program unsustainable and unable to reach all of the fourth- and fifth-grade teachers in the district. This study supports the notion that leadership tasks are distributed throughout the school system, and thus the capacity for reform is not dictated only by discrete program features [29]. In other words, the existence of strong multi-institutional partnerships and high-quality curricula and teacher PD was not enough to sustain the program. Rather it is the interaction of multiple features at various system levels that must be considered when developing the capacity for reform.

### 4.3. Conclusions

The overall goal of this program was to establish a strong, sustainable model-based inquiry program in the district that would become a regional model and resource for promoting students' interest and capacities to pursue careers in STEM fields. It is possible that the usability gaps that emerged could have been overcome if Rich and Sarah had more time and funds (beyond the 3-year funding cycle) to build on the district's capacity for change, as this condition is a well-known barrier to the success of educational reform

efforts [1,3,25,33]. And while there is evidence of short-term student gains, an important limitation to this study is that it is unknown whether the program influenced students' achievement and interest in STEM in any long-term sense. It could be that ambitious reform efforts, such as this one, require longer funding cycles to be both well understood (in terms of student impact) and to become sustainable. Future research pursuits should incorporate plans for the longitudinal monitoring of student participation outcomes in such programs in order to better understand whether the intended effects of increased long-term interest in STEM are realized.

Another limitation of this investigation involves only having interviewed a small subset of teachers (6 of 37, or 16%) who participated in the program. It is possible that teachers whose perspectives were not included in this study experienced the usability of the program in different ways. Additionally, we were unable to interview administrators in the district and so their unique perspectives were absent from this analysis. In order to follow DBIR and improvement science approaches towards system-level reform with greater fidelity, it would be essential to understand their perceived issues in the usability of the program as well. Additionally, and lastly, with regard to the study limitations, while we elucidated multiple issues that reformers may need to contend with in doing this important work, our investigation stops short of working to reduce the usability gaps we uncovered by tailoring the program features in ways that would allow for sustainable and scalable implementation within this district.

This study demonstrates some of the investigatory resources (i.e., incorporating the DBIR and improvement science frameworks to enact change and then evaluating outcomes using the usability cube framework) that can be used to understand the system variables that will clarify the important question of "what works, for whom, and under what conditions?". Research such as this, which contributes to a better understanding of the systematic variables affecting reform efforts, is critical to increasing the success of future efforts and is urgently needed in STEM subjects at the elementary level [1–3,6]. This study further highlights that accomplishing the type of change need throughout all the system levels is extremely complex and difficult to achieve, and curricular innovations often do not last beyond the presence of the innovators, as was the case here [1,12]. As Rich put it, "I'm not sure that we're big enough [at the ecology center] to create the kind of change that the world needs for [this] type of work." However, it is known that the relative success of these efforts is deeply contextualized and, given the diversity of educational systems, there is no single approach that can predictably lead to change [1,12,25]. Therefore, it is important for innovators to continue engaging in the formidable task of reform in order for the field as whole to gain insight into and learn from the collective experiences of these efforts.

**Author Contributions:** Conceptualization, S.A.Y., B.C. and A.M.C.; methodology, A.M.C., B.C. and J.S.; validation, A.M.C., S.A.Y., B.C., S.C. and J.S.; formal analysis, A.M.C.; writing—original draft preparation, A.M.C.; writing—review and editing, S.A.Y., B.C., J.S. and S.C.; project administration, B.C. and S.C.; funding acquisition, B.C. All authors have read and agreed to the published version of the manuscript.

**Funding:** This research was largely funded by the National Science Foundation, grant number 1513043, and, in part, by the Litzsinger Road Ecology Foundation.

**Institutional Review Board Statement:** This study was conducted according to the federal guidelines for Human Subjects Research in the U.S.A. and approved by the Institutional Review Board of the University of Pennsylvania (protocol #831918 approved on 31 October 2018).

**Informed Consent Statement:** Informed consent was obtained from all subjects involved in the study.

**Data Availability Statement:** Consistent with the ethical guidelines of Human Subjects Research, all data collected for this project are accessible only to faculty, students and staff who have obtained IRB training and approval as per the University of Pennsylvania's IRB regulations and have been assigned to the project.

**Acknowledgments:** The authors would like to thank the developers of the modeling tool and other researchers who have been involved in various aspects of this work, including Daniel Wendel, Eric Klopfer, and Irene Lee.

**Conflicts of Interest:** The authors declare no conflict of interest. The funders had no role in the design of the study; in the collection, analyses, or interpretation of data; in the writing of the manuscript; or in the decision to publish the results.

## Appendix A

All of the model-based inquiry lessons developed as part of this program are archived and accessible on the Project GUTS (Growing Up Thinking Scientifically) website. In order to access the material, you must register for a free account through the Project GUTS website. The direct links to the six modules developed as part of this study are included below:

Module 1—Introduction to Modeling and Simulation:

https://teacherswithguts.org/resources/modelbest-module-1-introduction-to-modeling-and-simulation

Module 2—Catching Butterflies:

https://teacherswithguts.org/resources/modelbest-module-2-catching-butterflies

Module 3—Predator/Prey Survival Behaviors:

https://teacherswithguts.org/resources/modelbest-module-3-predator-prey-survival-behaviors

Module 4—Simple Garden Ecosystem:

https://teacherswithguts.org/resources/modelbest-module-4-simple-garden-ecosystem

Module 5—Water Pollution:

https://teacherswithguts.org/resources/modelbest-module-5-water-pollution

Module 6—Air Pollution:

https://teacherswithguts.org/resources/modelbest-module-6-air-pollution

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
