# Peer review of "Building System Capacity with a Modeling-Based Inquiry Program for Elementary Students: A Case Study"

_systems, doi:10.3390/systems9010009_

Round 1
Reviewer 1 Report
It is a good paper and I would like to enhance the authors to complement with some papers that reflect conducted studies. Related to scientific literature about instructional strategies as for accomplish alternatives to traditional pedagogy, the authors express in lines 71 to 76: “Model-based learning refers to the understanding gained from the ability to create, manipulate, and communicate models; to the extent that model-based learning involves being able to collect and use data, it overlaps with the skills needed for data literacy [18]. As an instructional strategy, model based learning offers a compelling alternative to traditional pedagogy as well as an opportunity for elementary students to use and engage with sense-making tools that are pervasive in modern science [19].” I recommend the following study: Camerino O, Valero-Valenzuela A, Prat Q, Manzano Sánchez D, and Castañer, M. (2019) Optimizing Education: A Mixed Methods Approach Oriented to Teaching Personal and Social Responsibility (TPSR). Frontiers in. Psychology. 10:1439. http://dx.doi.org/10.3389/fpsyg.2019.01439
In line 731, the authors mention student’s motivation but it is strange that they not include this aspect that underpins the innovation of science education.
In line 228 the authors refer to: “…teachers represented a broad range of teaching experience (from approximately 1 to 10 years of experience).”It could be of interest to refer to some literature that exposes the criteria of novice and expert teachers
Related to the methodology I didn’t found to crucial aspects:
1) the ethical committee that had consent the study;
2) the validation of the interview.
I would like to know if both aspects have been taken into account with the respective evidences about how they have been implemented.
Author Response
1. I recommend the following study: Camerino O, Valero-Valenzuela A, Prat Q, Manzano Sánchez D, and Castañer, M. (2019) Optimizing Education: A Mixed Methods Approach Oriented to Teaching Personal and Social Responsibility (TPSR). Frontiers in. Psychology. 10:1439. http://dx.doi.org/10.3389/fpsyg.2019.01439
This recommended paper describes a distinct and unrelated teaching approach that is not relevant to the model-based learning approach described here. Unfortunately, we do not agree that this paper is connected to the pedagogical approach of model-based learning that we utilize and reference. Thank you for your recommendation.
2. In line 731, the authors mention student’s motivation but it is strange that they not include this aspect that underpins the innovation of science education.
Now line 787: We deleted the word “motivations” here and replaced it with “interest” since it seemed to be confusing.
3. In line 228 the authors refer to: “…teachers represented a broad range of teaching experience (from approximately 1 to 10 years of experience).”It could be of interest to refer to some literature that exposes the criteria of novice and expert teachers.
To this we added (now line 256-258): “the six fifth-grade teachers represented a broad range of teaching experience (from approximately 1 to 10 years of experience). This allowed us to sample various perspectives concerning implementing model-based inquiry in the classrooms, as teachers with more experience have been found to be more proficient than novices in accomplishing teaching goals and in enacting high-quality instruction (Jegede et al., 2000; Yoon et al., 2019).
Related to the methodology I didn’t found to crucial aspects:
1) the ethical committee that had consent the study; This is referenced on line 200 as: “Institutional Review Board approved protocol #831918”.
2) the validation of the interview.
To address this concern, we clarified our validation process in line 295-313:
“The study questions, framework, and data-collection techniques were constructed so as to maintain as naturalistic a methodology as possible [48]. The interview data went through a systematic process of validation. First, we used a modified grounded theory approach in evaluating this data source, where we searched for evidence of the a priori categories related to student affordances and the usability cube dimensions (i.e., capability, school culture, and policy/management) as either positive features (i.e., decreased the usability gap) or negative features of the system (i.e., increased the usability gap) [49]. Using a constant comparative method of analysis, the first author read the first interview where information that related to each of those categories was derived, and that information was then compared to and triangulated with each subsequent interview in order to validate that a particular finding emerged from multiple sources [49]. We also triangulated the evidence that emerged across the separate perspectives of the teachers, Rich, and Sarah. Gathering interview data from both the participant teachers and the program coordinators represented emic (or within social group) and etic (or outside social group) perspectives, which allowed for a more holistic picture of the supports and challenges that teachers encountered in their program participation [50]. The second systematic way we validated the findings was to work with authors 2 and 4 in order to negotiate and resolve any discrepancies in categorization [51]. And third, we used member checking in which we validated the findings with two of the interviewees that were available, and this helped to ensure accuracy and validity in our representation of the findings [50].”
Reviewer 2 Report
The authors present a work that, although it may be more appropriate for a journal focused on educational issues, could also be interesting for the journal´s readers.
The manuscript is well structured, addresses in depth the issue under study, and is easily readable.
Having said this, even though, in general terms, both the manuscript and the underlying research are adequate; there are some points that can be improved.
- Methodology
The authors develop this section with a high level of detail. However, it would be interesting to add additional bibliographic references to justify some of the methodological decisions made during the research. Especially as regards sections 2.1. Context and 2.2. Participants.
In addition, regarding the section 2.2. Participants, although sociodemographic data on the population are given, such information is not provided for the sample. On the other hand, the significance of such a sample is not discussed either. In this respect of significance, in case is not possible to justify that issue, it would be convenient to acknowledge it openly.
- Results
The results provided are rich and interesting. However, in reviewer’s opinion, it may be interesting to add more teachers’ testimonials.
- Discussion
The content in the discussion is adequate and provides appropriate references. This section also includes a subsection on "4.3 Concluding Thoughts ...". However, it would be interesting to complement this final part of the manuscript with a subsection of limitations (the common ones of qualitative study like the one in question) and future lines of research.
Author Response
- Methodology
1. The authors develop this section with a high level of detail. However, it would be interesting to add additional bibliographic references to justify some of the methodological decisions made during the research. Especially as regards sections 2.1. Context and 2.2. Participants.
Added to line 216: StarLogo Nova. http://www.slnova.org/ retrieved Jan. 2020
Resnick, M. (1994). Turtles, termites, and traffic jams: Explorations in massively parallel microworlds. Cambridge, MA: MIT Press.
Added to line 223-26:
Brand, B. R. (2020). Integrating science and engineering practices: Outcomes from a collaborative professional development. International Journal of STEM Education, 7(1), 1–13.
Dare, E. A., Ellis, J. A., & Roehrig, G. H. (2018). Understanding science teachers’ implementations of integrated STEM curricular units through a phenomenological multiple case study. International Journal of STEM Education, 5(1), 1–19.
Kelley, T. R., Knowles, J. G., Holland, J.D., & Jung, H. (2016). A conceptual framework for integrated STEM education. International Journal of STEM Education, 3(1), 1–11.
Added to line 243-246:
Darling-Hammond, L., Hyler, M. E., & Gardner, M. (2017), Effective Teacher Professional Development. Learning Policy Institute.
Desimone, L. (2009). Improving impact studies of teachers’ professional development: Toward better conceptualizations and measures. Educational Researcher, 38(4), 181–199.
Desimone, L. M., & Garet, M. S. (2015). Best practices in teachers’ professional development in the United States. Psychology, Society and Education, 7(3), 252–263.
Added to line 256-57:
Jegede, O., Taplin, M., Chan, S. Trainee teachers' perception of their knowledge about expert teaching. Educational Research 2000, 42, 287–308.
Yoon, S., Evans, C., Anderson, E., Koehler, J., Miller, K. Validating a model for assessing science teacher’s adaptive expertise with computer-supported complex systems curricula and its relationship to student learning outcomes. Journal of Science Teacher Education 2019, 30, 890–905.
2. In addition, regarding the section 2.2. Participants, although sociodemographic data on the population are given, such information is not provided for the sample. On the other hand, the significance of such a sample is not discussed either. In this respect of significance, in case is not possible to justify that issue, it would be convenient to acknowledge it openly.
On line 265-269 these data are provided on the sample of students who were video-observed.
The 2nd point, we added statement addressing the significance of the sample (line 260-62: “Therefore, the sociodemographic composition of this district illustrates that this reform effort was targeted towards a population of students that is in critical need of improved educational resources in STEM subjects [1,5,10, 24].
Results
3.The results provided are rich and interesting. However, in reviewer’s opinion, it may be interesting to add more teachers’ testimonials.
Added more testimonial here: line 387-388:“This really amped up [student] knowledge and really I think it checked off all of the markers of the standards we are teaching right now”
Line 400-01: Teacher 6 added, “I really think that how you guys laid out the lesson plans, that was amazing…when it’s already built in as lessons, then it’s pretty much there for us [to use].”
Line 493-495: Teacher 2 confirmed that “the scheduling was just difficult because we didn’t really touch base with scientific things until we did [this curriculum]. . . I think if we had a set schedule where science was implemented throughout the Fall it would’ve been easier.”
Line 545-548: Teacher 1 added that “I think that [this curriculum] is what this generation is going to be doing in the future. . . A lot of them are going to have to do lessons online in the future, so I do think that this helps prepare our kids for the future better than the curriculum I have now.”
Line 655-657: Teacher 5 confirmed that “[the curriculum] really gets the kids interested and it can reinforce the things that they’re learning. . . I enjoyed it, my kids really enjoyed it, I think they’re getting a lot out of it.”
Discussion
4. The content in the discussion is adequate and provides appropriate references. This section also includes a subsection on "4.3 Concluding Thoughts ...". However, it would be interesting to complement this final part of the manuscript with a subsection of limitations (the common ones of qualitative study like the one in question) and future lines of research.
We revised section 4.3 to address the limitations and future directions of this study. Now lines784-822.
“4.3 Conclusions
The overall goal of this program was to establish a strong, sustainable model-based inquiry program in the district that would become a regional model and resource for promoting students’ interest and capacities to pursue careers in STEM fields. It is possible that the usability gaps that emerged could have been overcome if Rich and Sarah had more time and funds (beyond the 3-year funding cycle) to build on the district’s capacity for change, as this condition is a well-known barrier to the success of educational reform efforts [1,3,25,33]. And while there is evidence of short-term student gains, an important limitation to this study is that it is unknown whether the program influenced students’ achievement and interest in STEM in any long-term sense. It could be that ambitious reform efforts, such as this one, require longer funding cycles to be both well understood (in terms of student impact) and to become sustainable. Future research pursuits should incorporate plans for longitudinal monitoring of student participation outcomes in such programs in order to better understand whether the intended effects of increased long-term interest in STEM are realized.
Another limitation of this investigation involves only having interviewed a small subset of teachers (6 of 37, or 16%) who participated in the program. It is possible that teachers whose perspectives were not included in this study experienced the usability of the program in different ways. Additionally, we were unable to interview administrators in the district and so their unique perspectives were absent from this analysis. In order to follow DBIR and improvement science approaches towards system-level reform with greater fidelity, it would be essential to understand their perceived issues in the usability of the program as well. And lastly with regards to study limitations, while we elucidated multiple issues that reformers may need to contend with in doing this important work, our investigation stops short of working to reduce the usability gaps we uncovered by tailoring the program features in ways that would allow for sustainable and scalable implementation within this district.
This study demonstrates some of the investigatory resources (i.e., incorporating the DBIR and improvement science frameworks to enact change and then evaluating outcomes using the usability cube framework) that can be used to understand the system variables that will clarify the important question of “what works, for whom, and under what conditions”. Research such as this, which contributes to a better understanding the systematic variables affecting reform efforts, is critical to increasing the success of future efforts and is urgently needed in STEM subjects at the elementary level [1–3, 6]. This study further highlights that accomplishing the type of change need throughout all the system levels is extremely complex and difficult to achieve, and curricular innovations often do not last beyond the presence of the innovators, as was the case here [1,12]. As Rich put it, “I'm not sure that we're big enough [at the ecology center] to create the kind of change that the world needs for [this] type of work.” However, it is known that the relative success of these efforts is deeply contextualized and, given the diversity of educational systems, there is no single approach that can predictably lead to change [1,12,25]. Therefore, it is important for innovators to continue engaging in the formidable task of reform in order for the field as whole to gain insight into and learn from the collective experiences of these efforts.”
We also suggest another interesting pursuit for future research on 756-59: “To help close this usability gap, future programs aimed at reforming elementary science might consider aligning activities more explicitly with math and literacy standards and reimagine the reform effort as a means to contextualize these higher priority subjects into a more integrative STEM education approach [33,53,61]. “
Reviewer 3 Report
Paper deal with an interesting problem where experimental investigation is presented and evaluated.
The main problem is in the absence of theoretical background with literature on which the theory is base. No comparison to other work in the field. No clear contribution, no conclusions..
Also the need for this investigation is not clear!
Paper has a low potential in current version. SO for next iteration please update at minimum:
1) references need to be focused to research topic and problems dealt in paper ..
2) Not only Introduction or related work section need to be supported by references.. whole paper need to be based on previous findings.. Number of references is low.
3) Need of comparative evaluation is in these papers more than welcome to have a possibility to compare relevant findings or same style of testing..
4) methodology need to be enhanced.. current is a little bit unbalanced - superficial..
5) there is lack of design description.. no detailed info.. only abstract level.. authors need to go more in details..
6) no clear contribution. it nww to be clear from abstarct as wella s from conclusions what is the pros and cons..
7) Adding some papers published in recent years in this journal is very mandatory. How you would confirm closenes to journal topics? I found only one..
Author Response
1. The main problem is in the absence of theoretical background with literature on which the theory is base. No comparison to other work in the field. No clear contribution, no conclusions..
The theoretical framework we outline is that we incorporate the DBIR and improvement science frameworks to enact a STEM education reform and then evaluate outcomes using the usability cube framework.
4.3 changed “Concluding thoughts on the Challenges and Complexities of Reform Efforts” to “Conclusions” and extended this section to more clearly signal the conclusions of the paper and elaborate on what they are. (Now line 784-822)
We reworded and added to the contributions of the work in the conclusions to make the intended contribution of this work more clear (line 808-814): “This study demonstrates some of the investigatory resources (i.e., incorporating the DBIR and improvement science frameworks to enact change and then evaluating outcomes using the usability cube framework) that can be used to understand the system variables that will clarify the important question of “what works, for whom, and under what conditions”. Research such as this, which contributes to a better understanding the systematic variables affecting reform efforts, is critical to increasing the success of future efforts and is urgently needed in STEM subjects at the elementary level [1–3, 6].”
2. Also the need for this investigation is not clear!
We highlight the need for this investigation at several points during the introduction.
Line 30-33: “The need for this approach has never been truer in science, technology, engineering, and mathematical (STEM) education, where continued workforce shortages and underrepresentation of women and minorities in STEM occupations in the United States necessitate effective, long-lasting, and extensive science education reform [4,5].”
Line 95-97: “There is a clear need to focus educational research and innovation efforts at the system-level in order to develop effective and long-standing adoption of curricula that can support data literacy skills and model-based learning for all young learners [26].”
Line 123-127: “In other words, establishing a research base that clarifies challenges in implementing reform across a range of contexts can equip others involved in science education reform with an understanding of how similar innovations could be adapted in their district, thereby addressing the need to generate more knowledge about “what works, for whom, and under what conditions?”
Line 145-46:” In this study we employed the DBIR and aspects of the improvement science frameworks to address the need to bring widespread innovative and effective STEM education to elementary students [23].”
1. references need to be focused to research topic and problems dealt in paper ..
The references in the paper were cited because they provide the framework for and contribute ideas to the topics discussed throughout the paper. Thank you for your advice, we add some references.
2) Not only Introduction or related work section need to be supported by references.. whole paper need to be based on previous findings.. Number of references is low.
We added 16 more references and references are found in all sections of the paper (but to a much lesser extent in the results section, where our main intent is to present the results of our case study).
3.Need of comparative evaluation is in these papers more than welcome to have a possibility to compare relevant findings or same style of testing..
Since this is a non-comparative qualitative case study (line 192-194) that suggests a framing for understanding educational reform, we do not see this lack of comparison as a limitation. We understand that researchers from other methodological traditions might be more interested in a more comparative approach, which this is not.
4. methodology need to be enhanced.. current is a little bit unbalanced - superficial..
We added more detail throughout the methods section to address this. Most significantly, we clarified our validation process in line 295-313.
5) there is lack of design description.. no detailed info.. only abstract level.. authors need to go more in details..
We added more references and text in order to support and clarify the design decisions specified in the methods.
line 223-27: “Addressing these features through carefully designed PD was important since it is known that teachers often exhibit low confidence in teaching science content without formal training and can also find it difficult to teach new, innovative pedagogical approaches without practical advice and exposure in how to teach it [40–42].”
Line 243-246: “In short, this PD included aspects such as 1) modeling effective instruction, 2) offering coaching and expert support, 3) dedicating time for feedback and reflection on practice and 4) sustained duration of PD participation during the academic year. These aspects are known to be crucial in engaging teachers in effective and high-quality PD [43–45].”
Line 255-58: “This allowed us to sample various perspectives concerning implementing model-based inquiry in the classrooms, as teachers with more experience have been found to be more proficient than novices in accomplishing teaching goals and in enacting high-quality instruction [46,47].”
Line 260-62: “Therefore, the sociodemographic composition of this district illustrates that this reform effort was targeted towards a population of students that is in critical need of improved educational resources in STEM subjects [1,5,10, 24].”
Line 295-313: “The study questions, framework, and data-collection techniques were constructed so as to maintain as naturalistic a methodology as possible [48]. The interview data went through a systematic process of validation. First, we used a modified grounded theory approach in evaluating this data source, where we searched for evidence of the a priori categories related to student affordances and the usability cube dimensions (i.e., capability, school culture, and policy/management) as either positive features (i.e., decreased the usability gap) or negative features of the system (i.e., increased the usability gap) [49]. Using a constant comparative method of analysis, the first author read the first interview where information that related to each of those categories was derived, and that information was then compared to and triangulated with each subsequent interview in order to validate that a particular finding emerged from multiple sources [49]. We also triangulated the evidence that emerged across the separate perspectives of the teachers, Rich, and Sarah. Gathering interview data from both the participant teachers and the program coordinators represented emic (or within social group) and etic (or outside social group) perspectives, which allowed for a more holistic picture of the supports and challenges that teachers encountered in their program participation [50]. The second systematic way we validated the findings was to work with authors 2 and 4 in order to negotiate and resolve any discrepancies in categorization [51]. And third, we used member checking in which we validated the findings with two of the interviewees that were available, and this helped to ensure accuracy and validity in our representation of the findings [50]. “
6) n clear contribution. it nww to be clear from abstarct as wella s from conclusions what is the pros and cons.
We make a general statement addressing the contribution in the abstract with line 20-23: “In light of these findings, we discuss broader implications for building the capacity to overcome system barriers. In this way, an in-depth examination of the context-specific barriers to reform in this educational system can inform efforts for future reform and innovation design.”
Line 46-52:” In this way, we intend to contribute to the corpus of research that aims to identify the practices and features needed for sustainability (i.e., existing beyond the time the researcher is in the classroom) and scalability (i.e., spreading beyond participating teachers and schools) of important early science education reforms. Additionally, we intend to demonstrate the investigatory resources that can be used to understand the system-level variables impacting the success of a reform effort, which in turn will help us tailor future reform efforts to local conditions.”
In the conclusion, section 4.3 changed from “Concluding thoughts on the Challenges and Complexities of Reform Efforts” to “Conclusions” and extended this section to more clearly signal the conclusions of the paper and elaborate on what they are. (Now line 784-822)
7) Adding some papers published in recent years in this journal is very mandatory. How you would confirm closenes to journal topics? I found only one..
Added two papers into the discussion (line 724-28) from the journal:
Pierson, A.E., Brady, C.E. Expanding opportunities for systems thinking, conceptual learning, and participation through embodied and computational modeling. Systems 2020, 8, 48.
Haas, A., Grapin, S.E., Wendel, D., Llosa, L., Lee, O. How fifth-grade English learners engage in systems thinking using computational models. Systems 2020, 8, 47.
Round 2
Reviewer 1 Report
The paper in the present form is now OK.
Reviewer 2 Report
In reviewer's opinion, the three issues mentioned during the first round of reviews (methodology, results, and discussion / conclusions) have been adequately addressed by the authors.
The methodology has improved in its rigor. The testimonials incorporated into the results section enrich the manuscript's reading. And finally, the discussion / conclusions part includes several reflections on the limitations of the study and its future lines of research.
However, on this last point, I must insist in my suggestion to present this part of limitations and future research in a separate subheading. Currently, the authors present this part within the conclusions heading, which leads the reader to understand that study's conclusions are mainly a bunch of limitations.
Considering all the above, we have a much more solid, rich, and rigorous manuscript.
Reviewer 3 Report
Dear Authors,
Thanks for all the corrections. I am impressed with the extent of the changes you have made to the manuscript. Overall, the quality of the article has increased significantly.
Authors incorporated my suggestion a most in the sense of updated references, comparation, highligting pros and cons and deeper description of methods.
Thus I suggest to accept article in present form.